# Dietary Macronutrient Intake and Cardiovascular Disease Risk and Mortality: A Systematic Review and Dose-Response Meta-Analysis of Prospective Cohort Studies

**DOI:** 10.3390/nu16010152

**Published:** 2024-01-02

**Authors:** Yibin Ma, Zekun Zheng, Litao Zhuang, Huiting Wang, Anni Li, Liangkai Chen, Liegang Liu

**Affiliations:** 1Department of Nutrition and Food Hygiene, Hubei Key Laboratory of Food Nutrition and Safety, School of Public Health, Tongji Medical College, Huazhong University of Science and Technology, Wuhan 430030, China; myb@hust.edu.cn (Y.M.); m202275496@hust.edu.cn (Z.Z.); clk@hust.edu.cn (L.C.); 2Ministry of Education Key Lab of Environment and Health, School of Public Health, Tongji Medical College, Huazhong University of Science and Technology, Wuhan 430030, China; zlt_29@163.com (L.Z.); 22211020023@m.fudan.edu.cn (H.W.); lianni_hust@163.com (A.L.)

**Keywords:** macronutrient intake, cardiovascular disease, mortality, meta-analysis

## Abstract

Many epidemiological studies have evaluated the intake of macronutrients and the risk of mortality and cardiovascular disease (CVD). However, current evidence is conflicting and warrants further investigation. Therefore, we carried out an umbrella review to examine and quantify the potential dose-response association of dietary macronutrient intake with CVD morbidity and mortality. Prospective cohort studies from PubMed, Embase, and CENTRAL were reviewed, which reported associations of macronutrients (protein, fat, and carbohydrate) with all-cause, CVD, cancer mortality, or CVD events. Multivariable relative risks (RR) were pooled, and heterogeneity was assessed. The results of 124 prospective cohort studies were included in the systematic review and 101 in the meta-analysis. During the follow-up period from 2.2 to 30 years, 506,086 deaths and 79,585 CVD events occurred among 5,107,821 participants. High total protein intake was associated with low CVD morbidity (RR 0.88, 95% confidence interval 0.82–0.94), while high total carbohydrate intake was associated with high CVD morbidity (1.08, 1.02–1.13). For fats, a high intake of total fat was associated with a decreased all-cause mortality risk (0.92, 0.85–0.99). Saturated fatty acid intake was only associated with cancer mortality (1.10, 1.06–1.14); Both monounsaturated fatty acid (MUFA) and polyunsaturated fatty acids (PUFA) intake was associated with all-cause mortality (MUFA: 0.92, 0.86–0.98; PUFA: 0.91, 0.86–0.96). This meta-analysis supports that protein intake is associated with a decreased risk of CVD morbidity, while carbohydrate intake is associated with an increased risk of CVD morbidity. High total fat intake is associated with a low risk of all-cause mortality, and this effect was different in an analysis stratified by the type of fat.

## 1. Introduction

Cardiovascular disease (CVD) remains the main cause of global mortality [1]. A well-balanced diet plays a significant role in the prevention of CVD and mortality, and dietary nutrient intake is associated with the occurrence, development, and treatment of CVD morbidity and mortality [2,3,4,5]. Carbohydrates, fats, proteins, minerals, vitamins, dietary fiber, and water are the nutrients needed by the human body. Carbohydrates, fats, and proteins, owing to their substantial requirements, constitute a significant portion of the diet and are termed as macronutrients. The current dietary guidelines advise against high-fat diets, particularly reducing the intake of saturated fatty acids (SFA) [6]. Furthermore, foods rich in protein are also considered as core components of a healthy eating pattern [6]. The World Health Organization’s guidelines suggest reducing total fat intake in exchange for a higher intake of carbohydrates [7]. However, the long-term effects of macronutrients on health outcomes are both incomplete and contradictory. Meta-analyses, encompassing several large-scale cohort studies from North America and Europe, indicate a correlation between elevated mortality rates and reduced carbohydrate intake [8,9,10,11,12]. A multi-national prospective study published in *The Lancet* in 2017 revealed that an increase in carbohydrate intake is associated with an increase in all-cause mortality, while individuals with lower all-cause mortality have higher intakes of total fat, monounsaturated fat (MUFA), polyunsaturated fat (PUFA), and SFA [13]. Additionally, a recent meta-analysis suggests that both an excessively high and low proportion of carbohydrate diets are associated with heightened mortality risk, with the lowest risk observed when carbohydrate intake is between 50 and 55% [14]. A recent meta-analysis report discovered that higher overall protein intake is correlated with lower all-cause mortality, and the benefits are primarily attributed to the plant protein intake [15], yet other studies have been unable to draw similar conclusions [16,17,18,19]. The above research suggests that the traditional recommended dietary structure may need to be reconsidered. Therefore, we conducted a systematic review and dose-response meta-analysis of prospective cohort studies to summarize the association of dietary macronutrient intake with all-cause and cause-specific mortality and CVD morbidity.

## 2. Materials and Methods

This systematic review and meta-analysis were performed following the recommendation of the Cochrane Collaboration approach for Systematic Reviews [20]. We drafted the meta-analysis following the Preferred Reporting Items for Systematic Reviews and Meta-Analyses (PRISMA) [21]. Systematic review registration: PROSPERO CRD42021297059. All prospective cohort studies were included.

### 2.1. Search Methods for Identification of Studies

We systematically searched all articles published on the online database as of 20 March 2023, including PubMed, Embase, and Cochrane libraries, without language or publication time restrictions. To avoid publication loss, the reference list of retrieved articles, as well as previous systems and narrative comments, were manually searched.

### 2.2. Eligibility Criteria for Study Selection

Duplicate publications were deleted. After reviewing the titles and abstracts, the two authors individually assessed and evaluated each study to decide whether it should be included. Any disagreement was resolved through discussions and consensus among all authors. If the study met all of the following criteria, it was included in this meta-analysis: (1) prospective cohort study, (2) reported effect sizes including hazard ratios (HR) or relative risks (RR) with the corresponding 95% confidence intervals (CIs), (3) reported data on at least one exposure of interest, (4) reported data on at least one outcome of interest. Uncontrolled case reports, animal studies, letters, comments, reviews, meta-analyses, and ecological studies were excluded. Moreover, we excluded studies with insufficient data or performed on children (Appendix A).

### 2.3. Data Extraction and Management

The data were independently extracted by two authors and checked against each other. The extracted details included the study design, behavioral country, exposure and outcome evaluation, participant characteristics, and statistical analysis, including confounding factor adjustments included in the study using pretest instruments. If necessary, the author was contacted for further information.

### 2.4. Assessment of Risk of Bias in Included Studies

The risk of bias in the study was assessed in accordance with the domains recommended by the Newcastle Ottawa Scale (NOS), encompassing the selection of study groups, comparability among groups, and determination of exposure or outcomes [22,23]. Studies scoring at least 7 were categorized as high-quality (implying low risk of bias), studies scoring 6 were considered medium-quality, and those scoring ≤ 5 were deemed low-quality. Appendix A provides detailed information on the NOS scale.

### 2.5. Statistical Analysis

RR and HR (as well as 95% CI) were employed to compare the highest and lowest categories of intake for protein, fat, and carbohydrates to calculate the standard errors of the log RR and HR. We used a random-effects model to minimize potential heterogeneity. *I*^2^ was used to statistically evaluate the degree of heterogeneity between experiments, with a value of 50% or higher indicating a relatively high degree of heterogeneity [20,24]. If there was substantial heterogeneity among studies, a subgroup analysis was performed to investigate potential sources of heterogeneity.

We performed a priori sensitivity analyses by removing every single study from the meta-analyses and recalculating the summary association (the “leave one out” approach) [24]. If over 10 studies were available, reporting bias was assessed using the Egger’s and Begg’s tests. To explore the possible nonlinear association between dietary macronutrient intake and related outcomes, we used a two-stage random effect dose-response meta-analysis model. Statistical analyses were conducted using STATA version 15.1. A two-sided *p*-value of less than 0.05 was considered significant for all tests.

## 3. Results

### 3.1. Literature Search

Overall, 60,544 articles were found by searching the PubMed, EMBASE, and Cochrane libraries. After eliminating duplicates and publications that did not meet the criteria for inclusion, 875 potential full-text articles relevant to this study were identified. Through full-text review, we excluded an additional 753 articles (Figure 1). Lastly, 124 cohort study reports were incorporated into the systematic review, and 101 were subsequently included in the meta-analysis. Among the studies related to mortality, 40 reported on the impact of protein intake, 46 reported on fat intake, and 19 reported on carbohydrate intake. Regarding CVD events, we identified 8 publications for protein, 44 for fat, and 9 for carbohydrates.

### 3.2. Characteristics of Included Studies

Table 1 summarizes the characteristics of the included studies. Appendix A show the characteristics of the included prospective cohort studies. The participants in these studies varied in number from 227 to 521,120, and their ages spanned from 19 to 99 years old. A total of 124 studies including 107,821 participants were included in this systematic review. Over the 2.2–30-year follow-up period, there were 79,585 cardiovascular events, of which 21,259 were strokes. Additionally, 506,086 deaths were recorded, of which 95,497 were due to cardiovascular diseases and 84,521 were due to cancers. Moreover, 18 articles included only men, and 27 publications included only women. Of the remaining studies, 22 publications reported RR for men and women separately. In total, 58 publications described studies in North America, 33 in Europe, 29 in Asia, 2 in Australia, 1 in South America, and 1 in the populations from 18 different countries. The incidence rate and mortality rate were ascertained through medical records, death certificates, or equivalent evidence, or self-reporting. Diet was assessed using food frequency questionnaires, 24 h recalls, or food records. Appendix A describes the aspects of diet and outcome measurement for each group included study.

### 3.3. Study Quality Assessment

Appendix A present the covariate adjustments in the prospective cohort studies. The majority of the studies examined in the analysis controlled for energy intake, a predetermined primary confounding variable. Moreover, 124 studies adjusted for at least four out of six significant confounding variables: age, gender, socioeconomic status, other health conditions, and lifestyle factors such as smoking and physical activity.

Appendix A displays the summary of bias risk assessments for the included prospective cohort studies. Prospective cohort studies are of relatively high quality, with an NOS score ≥ 6, as determined by their risk of bias. Therefore, in most research comparisons, there is no overall concern about the risk of bias.

### 3.4. Systematic Review

#### 3.4.1. Systematic Review for All-Cause Mortality and Cause-Specific Mortality

Of 65 cohorts on the association between intake of protein and all-cause mortality, thirteen reported an inverse association (total protein: 6; plant protein: 7), while eight showed a positive association (total protein: 4; animal protein: 4). For the association between intake of fat and all-cause mortality, 30 studies showed an inverse association (total fat: 7; SFA: 4; MUFA: 8; PUFA: 7; ω-3 fatty acids: 1; linoleic acid: 3), and 22 reported a positive association (total fat: 1; SFA: 11; MUFA: 3; PUFA: 4; other fats: 3). In addition, 11 cohorts showed a positive correlation between carbohydrate intake and all-cause mortality, while 2 cohorts reported a negative correlation.

For CVD mortality, 11 cohorts reported an inverse association with intake of protein (total protein: 2; animal protein: 1; plant protein: 8), 21 cohorts with fat (total fat: 4; SFA: 3; MUFA: 6; PUFA: 6; other fats: 2), and 5 cohorts with carbohydrate. Moreover, 3 studies reported a positive association with protein intake (total protein: 1; animal protein: 2), 14 studies with fat intake (total fat: 3; SFA: 7; MUFA: 2; PUFA: 1), and 2 studies with carbohydrate intake.

Four cohorts indicated an inverse association between intake of protein and cancer mortality (total protein: 2; plant protein: 2), and four showed a positive association (total protein: 3; animal protein: 1). For the association between intake of fat and cancer mortality, five studies showed an inverse association (total fat: 3; SFA: 1), and four reported a positive association (total fat: 1; SFA: 10; MUFA: 2; PUFA: 3). In addition, three studies showed a positive association between carbohydrate intake and cancer mortality.

#### 3.4.2. Systematic Review for Cardiovascular Disease Events

Of 21 cohorts on the association between intake of protein and CVD events, 7 reported an inverse association (total protein: 3; animal protein: 2; soy protein: 2). For the association between intake of fat and CVD events, 40 studies showed an inverse association (total fat: 5; SFA: 16; MUFA: 6; PUFA: 6; plant fat: 2), and 10 reported a positive association (total fat: 3; SFA: 7). Moreover, four cohorts showed a positive association between carbohydrate intake and CVD events.

### 3.5. Meta-Analysis on Dietary Macronutrient Intake and CVD Morbidity and Mortality

#### 3.5.1. Associations of Protein Intake with All-Cause and Cause-Specific Mortality

Among 36 studies on total protein intake and mortality, 28 provided 59 cohorts for comparing the association between the highest and lowest categories of total protein intake in all-cause mortality (25), CVD mortality (17), and cancer mortality (17). Of 666,129 participants included in these articles, we documented 38,785 deaths, 21,155 CVD deaths, and 17,910 cancer deaths. No significant association was found between total protein intake and all-cause mortality. The summary effect size for comparing the highest and lowest intakes of total protein was 0.97 (95% CI: 0.93 to 1.02, *I*^2^ = 68.7%, *p* = 0.242). The same results were found in CVD mortality and cancer mortality. The effect size of the highest versus lowest intakes was 0.98, 95% CI: 0.94–1.03, *I*^2^ = 17.2%, *p* = 0.448 for CVD mortality; and RR: 0.97, 95% CI: 0.91–1.03, *I*^2^ = 58.3%, *p* = 0.267 for cancer mortality.

When examining the association between animal protein intake and mortality in 32 cohorts, comprising a total of 411,303 participants and 84,841 deaths, of which 23,281 were due to CVD and 30,518 due to cancer, we did not detect a significant correlation. The effect size of the highest versus lowest intakes was RR: 1.04, 95% CI: 0.97–1.12, *I*^2^ = 68.9%, *p* = 0.309 for all-cause mortality; RR: 1.07, 95% CI: 0.97–1.18, *I*^2^ = 42.1%, *p* = 0.183 for CVD mortality; and RR: 0.98, 95% CI: 0.92–1.05, *I*^2^ = 13.2%, *p* = 0.598 for cancer mortality.

Consumption of plant protein, which was examined in 37 cohorts with a total of 962,646 participants and 198,711 deaths, including 48,563 CVD deaths and 40,801 cancer deaths, was inversely associated with all-cause mortality (RR: 0.92, 95% CI: 0.87–0.98, *I*^2^ = 43.8%, *p* = 0.010) and CVD mortality (RR: 0.85, 95% CI: 0.78–0.93, *I*^2^ = 19.5%, *p* = 0.000) (Figure 2).

#### 3.5.2. Associations of Protein Intake with CVD Morbidity

Seven publications with 15 cohorts examined the association between total protein intake and CVD events. These studies included 286,075 participants and 10,278 CVD events, including 4899 stroke events. A high intake of total protein was generally negatively correlated with the prevalence rates of CVD and stroke (highest versus lowest, CVD morbidity: 0.90 (0.84, 0.97), *I*^2^ = 38.8%, *p* = 0.006; stroke morbidity: 0.84 (0.75, 0.95), *I*^2^ = 0.0%, *p* = 0.004).

The intake of animal and plant proteins was examined across five cohorts, involving a total of 212,582 participants and 4175 cases of CVD, yielding consistent results (animal protein, CVD morbidity: 0.77 (0.63, 0.92), *I*^2^ = 52.2%, *p* = 0.008; plant protein, CVD morbidity: 0.85 (0.77, 0.95), *I*^2^ = 0.0%, *p* = 0.003) (Figure 2).

#### 3.5.3. Associations of Fat Intake with All-Cause and Cause-Specific Mortality

Overall, 25 studies, comprising 51 cohorts, examined the association between total fat intake and mortality, involving 966,545 participants and 89,614 deaths: 28,701 due to cardiovascular disease, and 25,344 due to cancer. We observed an inverse association between total fat intake and all-cause mortality (RR: 0.92 95% CI: 0.85–0.99, *I*^2^ = 79.3%, *p* = 0.019). However, no significant association was found between CVD mortality (RR: 0.96, 95% CI: 0.90–1.02, *I*^2^ = 69.4%, *p* = 0.195) and cancer mortality (RR: 1.00, 95% CI: 0.89–1.12, *I*^2^ = 68.0%, *p* = 0.827).

The association between consumption of SFA and mortality was examined in 29 publications with 59 cohorts, which included 1,283,730 participants, 214,284 deaths, 26,515 CVD deaths, and 25,344 cancer deaths. Higher SFA intake was associated with higher cancer mortality (RR: 1.10, 95% CI: 1.06–1.14, *I*^2^ = 0.0%, *p* = 0.000). No significant association was found neither in all-cause mortality (RR: 1.05, 95% CI: 0.98–1.13, *I*^2^ = 93.6%, *p* = 0.148) nor CVD mortality (RR: 1.03, 95% CI: 0.98, 1.08, *I*^2^ = 64.9%, *p* = 0.303).

Regarding the consumption of MUFA, 24 articles examined with approximately 50 cohorts, encompassing 1,417,944 participants and 213,836 deaths. This included 25,465 deaths from CVD and 25,354 from cancer. It was found that MUFA consumption was inversely associated with mortality. The effect size of the highest versus lowest intakes was RR: 0.92, 95% CI: 0.86–98, *I*^2^ = 88.0%, *p* = 0.012 for all-cause mortality; RR: 0.89, 95% CI: 0.84, 0.95, *I*^2^ = 42.7%, *p* = 0.000 for CVD mortality; and RR: 0.95, 95% CI: 0.91–0.99, *I*^2^ = 15.1%, *p* = 0.028 for cancer mortality.

Based on 25 publications involving 1,465,962 participants and 219,928 deaths, including 25,742 deaths from CVD and 25,354 from cancer, there was a consistent association observed between PUFA consumption and mortality. The effect size of the highest versus lowest intakes was RR: 0.91, 95% CI: 0.86–0.96, *I*^2^ = 79.7%, *p* = 0.001 for all-cause mortality; RR: 0.92, 95% CI: 0.85–1.00, *I*^2^ = 81.8%, *p* = 0.044 for CVD mortality; and RR: 0.94, 95% CI: 0.89–1.00, *I*^2^ = 23.7%, *p* = 0.051 for cancer mortality (Figure 3).

#### 3.5.4. Associations of Fat Intake with CVD Morbidity

Overall, 24 publications with 40 cohorts examined the association between total fat intake and CVD events. These studies included 1,266,207 participants and 43,699 CVD events, including 16,128 stroke events. No significant association was found between total fat intake and CVD morbidity (RR: 0.98, 95% CI: 0.94–1.02, *I*^2^ = 11.9%, *p* = 0.275) and stroke morbidity (RR: 0.97, 95% CI: 0.91–1.03, *I*^2^ = 1.5%, *p* = 0.248).

In 31 publications, the relationship between SFA intake and CVD events was studied, involving 49 cohorts with 1,364,664 participants, 46,877 CVD events, and 14,761 stroke events. No significant association was found in CVD morbidity (RR: 0.96, 95% CI: 0.92–1.02, *I*^2^ = 30.0%, *p* = 0.166) and stroke morbidity (RR: 0.92, 95% CI: 0.82–1.02, *I*^2^ = 47.6%, *p* = 0.117).

For MUFA consumption, which was examined in 21 articles, about 34 cohorts with a total of 998,778 participants, 29,410 CVD events, and 10,169 stroke events, there was no significant association (CVD morbidity: 0.95 (0.89, 1.02), *I*^2^ = 45.8%, *p* = 0.164; stroke morbidity: 0.95 (0.86, 1.06), *I*^2^ = 0.30%, *p* = 0.376).

The same finding was obtained for PUFA consumption and CVD events based on 23 publications of about 36 cohorts with a total of 999,005 CVD events, including 10,169 stroke events (CVD morbidity: 0.97 (0.92, 1.02), *I*^2^ = 15.0%, *p* = 0.275; stroke morbidity: 0.94 (0.87, 1.02), *I*^2^ = 0.0%, *p* = 0.116) (Figure 3).

#### 3.5.5. Associations of Carbohydrates Intake with All-Cause and Cause-Specific Mortality

In 16 publications, data from 31 cohorts were available, allowing for a comparison between the highest and lowest categories of carbohydrate intake. Out of the 544,359 participants covered by these articles, we documented 60,117 deaths: 16,592 due to CVD, and 17,452 due to cancer. There was no significant correlation observed between carbohydrate intake and mortality. The effect size of the highest versus lowest intakes was RR: 1.05, 95% CI: 0.99–1.17, *I*^2^ = 80.5%, *p* = 0.083 for all-cause mortality; RR: 1.02, 95% CI: 0.95–1.10, *I*^2^ = 51.9%, *p* = 0.548 for CVD mortality; and RR: 1.07, 95% CI: 0.98–1.17, *I*^2^ = 77.4%, *p* = 0.115 for cancer mortality (Figure 4).

#### 3.5.6. Associations of Carbohydrates Intake with CVD Morbidity

For carbohydrates consumption, which was examined in nine articles, about 20 cohorts with 580,933 participants, and 27,940 CVD events, there was an indication of an overall direct association between substitution of carbohydrates and CVD morbidity (RR: 1.07, 95% CI:1.02–1.13, *I*^2^ = 25.6%, *p* = 0.009), but not between substitution of carbohydrates and stroke morbidity (RR: 1.11, 95% CI: 0.93–1.31, *I*^2^ = 29.8%, *p* = 0.249) (Figure 4).

### 3.6. Non-Linear Dose-Response Analysis

#### 3.6.1. Non-Linear Dose-Response Analysis with Mortality

The dose-response analysis included eight publications on the relationship between total protein intake and all-cause mortality. An inverse association was found (*p* = 0.003 for non-linearity). This was also the case for CVD mortality based on six cohorts (*p* < 0.001 for non-linearity). However, no significant non-linear association was found between total protein intake and cancer mortality based on eight articles (*p* = 0.55 for non-linearity). The data from the dose-response analysis of animal protein intake and all-cause mortality in five publications did not exhibit a clear nonlinear association (*p* = 0.79 for non-linearity). Four publications obtained the same finding for CVD mortality (*p* = 0.09) and cancer mortality (*p* = 0.38). A non-linear dose-response analysis of five publications showed no significant association between plant protein intake and all-cause mortality (*p* = 0.18). The same results were found in cancer mortality (*p* = 0.81) and CVD mortality (*p* = 0.54) (Figure 5).

In the non-linear dose-response analysis, eight publications examining the association between total fat intake and mortality were incorporated. No significant association of total protein intake with all-cause mortality (*p* = 0.39), cancer mortality (*p* = 0.39), and CVD mortality (*p* = 0.05) was observed. Non-linear dose-response analysis showed that intake of SFA was not associated with mortality (*p* = 0.92 for all-cause mortality; *p* = 0.39 for cancer mortality; *p* = 0.78 for CVD mortality). For MUFA consumption based on 10 publications, an inverse association between MUFA intake and all-cause mortality was found (*p* < 0.05). Moreover, a significant non-linear association was found between PUFA intake and mortality based on eight publications (*p* < 0.001 for non-linearity). No significant non-linear association was seen between MUFA intake and cancer mortality (*p* = 0.41) and CVD mortality (*p* = 0.07) based on five articles. The same finding was obtained for PUFA consumption and cancer mortality (*p* = 0.57) and CVD mortality (*p* = 0.36) (Figure 6).

#### 3.6.2. Non-Linear Dose-Response Analysis with CVD Morbidity

As for carbohydrate consumption, four articles about seven cohorts were studied, and there was no significant nonlinear correlation between carbohydrate intake and the incidence rate of cardiovascular disease (*p* = 0.46). Among the six publications on the relationship between total protein intake and CVD events, three were included in the non-linear dose-response analysis. No significant association was found between total protein intake and CVD morbidity (*p* = 0.94). Non-linear dose-response analysis showed that total fat intake was not associated with CVD morbidity based on 10 publications (*p* = 0.55). This was also the case for SFA (*p* = 0.47; 11 publications), MUFA (*p* = 0.90; 8 publications), and PUFA (*p* = 0.96; 9 publications) (Figure 7).

### 3.7. Secondary Analyses and Publication Bias

#### 3.7.1. Subgroup Analyses

To ensure the robustness of the research results and delve into potential sources of heterogeneity among research, religion, and gender, subgroup analyses were performed according to predefined criteria, such as location and gender. Appendix A present findings for the different subgroups.

In studies performed in Europe, total protein intake was inversely associated with CVD morbidity in all females. The same finding was obtained between plant protein and all-cause mortality in women and North America. However, in terms of animal protein intake, there are indications that suggest an overall direct correlation between all-cause mortality in Europe’s studies and CVD mortality in North America’s studies.

A significant inverse association was observed between total fat intake and all-cause mortality in North American counties and in studies where the male-to-female ratio was less than 1. For PUFA intake, the same findings were also seen for all-cause and CVD mortality in women and North American counties. MUFA intake was inversely associated with CVD mortality in men and women in studies performed in North American and Europe countries. In addition, inverse associations between MUFA intake and CVD morbidity were observed in studies performed in Asia countries. Regarding SFA intake, there was an indication of an overall direct association with CVD mortality in North America. Nevertheless, a significant inverse association with CVD morbidity was seen in men and studies in Asia and Europe countries.

For carbohydrate consumption, a significant inverse association with all-cause mortality was seen in studies in North America and Europe countries. However, higher carbohydrate intake was associated with a higher CVD morbidity in women and studies performed in North America.

#### 3.7.2. Sensitivity Analysis

A priori sensitivity analyses removed every single study from the meta-analyses and recalculated the summary association (the “leave one out” approach), and the pooled RR remained unchanged. However, when conducting a sensitivity analysis using a random-effects model for total fat and cardiovascular disease mortality, the results shifted from a slightly significant inverse association to a substantial one after excluding the studies by Wakai et al. and Nagata et al. (RR: 0.85, 95% CI: 0.79–0.91, *I*^2^ = 10.8%, *p* = 0.023). Moreover, for animal protein and all-cause mortality, the removal of the study by Hernández-Alonso et al. and the study by Chen et al. shifted the overall estimate to 1.000 (0.950–1.053; *I*^2^ = 35.6% *p* = 0.144), suggesting these two studies were influential outliers. The same finding was obtained between SFA intake and all-cause mortality by removing Zhuang et al., Mazidi et al., and Tucker et al. (RR: 1.07, 95% CI: 1.03–1.12, *I*^2^ = 10.8%, *p* = 0.002).

#### 3.7.3. Publication Bias

In the meta-analysis, the Begg’s rank correlation test and Egger’s linear regression test were employed to assess potential publication bias. Apart from the total carbohydrate intake and all-cause mortality, no evidence of publication bias was identified in most studies using the Begg’s rank correlation test and Egger’s linear regression test. Trimming and filling methods did not alter the average effect size, indicating that publication bias did not affect the results (Appendix A).

## 4. Discussion

### 4.1. Principal Findings

Summarizing data from 124 prospective studies involving over five million participants, our research findings indicate that protein intake, including total protein, animal protein, and plant protein, is associated with a reduced risk of CVD, while carbohydrate intake is associated with an increased risk of CVD. Our research indicates that a high overall intake of fat is associated with a lower all-cause mortality rate; however, this association varies in analyses stratified by fat type. An increase in the intake of PUFA and MUFA is associated with a reduction in all-cause, CVD, and cancer mortality rates, while an increase in the intake of SFA is associated with an increase in cancer mortality rate.

### 4.2. Protein and Health Outcomes

We did not observe a correlation between total protein intake and mortality. Contrary to our observations, a meta-analysis [15] revealed that higher total protein intake is associated with a lower all-cause mortality. However, another meta-analysis indicated that there is a positive correlation between the overall intake of protein and the all-cause mortality [25]. The different results may be due to different protein sources.

Our research, along with other meta-analyses [15,25], suggests that plant protein intake is inversely associated with both all-cause and CVD mortality. Furthermore, legumes, grains, and nuts, serving as primary sources of plant-based protein, have been shown to contribute to a decrease in mortality from all-cause and CVD [26,27]. Animal protein is not associated with all-cause and cancer mortality. However, our results suggest a positive trend in its association with CVD mortality, which is consistent with previous systematic reviews and meta-analyses. The mechanism behind the association between protein and CVD mortality may be linked to the amino acid composition of the protein. Animal protein is rich in branched-chain amino acids and aromatic amino acids, whereas plant protein is not. High levels of branched-chain and aromatic amino acids are associated with insulin resistance and excess body weight, thereby increasing the risk of CVD mortality [28,29,30].

Our research indicates that a high intake of protein, from both animal and plant sources, may offer a protective effect on the incidence rate of CVD. Numerous studies have established the beneficial properties of various plant proteins [15,30]. Several studies [31,32] indicate that the higher the intake of red meat is, the higher the risk of CVD is; conversely, the higher the intake of poultry and fish is, the lower the risk of CVD disease is. This can explain the variations in types of animal protein and the associated risk of CVD disease.

In subgroup analysis, we observed opposite associations in the North American and Japanese populations, but not in the European population. This may be due to different sources of dietary plant protein in different populations. The primary source of plant protein in the diets of European populations is grains, and these findings may be associated with other non-protein dietary components also present in the entire food matrix. Consequently, it is imperative to meticulously characterize each type of plant and animal protein in order to precisely evaluate its association with CVD risk factors, incidence, and mortality.

### 4.3. Fat and Health Outcomes

Currently, there are significant differences between studies on fat intake and its relationship with all-cause and specific cause mortality [33,34,35,36,37,38,39,40,41,42,43,44,45,46,47]. Our study only found that SFAs intake was positively correlated with cancer mortality, but not all-cause mortality and CVD morbidity and mortality. However, a meta-analysis study indicated that high SFA is associated with elevated mortality from all-cause, CVD and cancer [48]. In contrast, another meta-analysis study did not find the above association [49]. Furthermore, a study revealed a notable correlation between SFAs intake and both all-cause and specific mortality in two study cohorts from NHANES [50]. However, studies examining SFAs and other primary causes of death, such as colon cancer [51] and various cancers [52,53], did not identify the aforementioned correlation. A study [54] indicated that the effect of dietary SFA on mortality may differ depending on its structure. This study [54] reported a strong negative correlation between intake of single-chain SFA and all-cause mortality. Upon the examination of SFA sources, the intake of processed and red meat is associated with increased mortality [55,56,57,58,59,60], whereas the intake of SFA from fish is associated with lower all-cause mortality [61]. Drawing from prior research outcomes and our meta-analysis, a higher mortality risk is associated with the total intake of SFA, yet the specific dietary sources and types of SFA might be the crucial determinants of this association. Due to the limited number of studies, we were unable to analyze according to the SFA subtype. It is more necessary to conduct specific characterization analysis on specific SFA subtypes and explore their impact on CVD.

Our findings regarding the inverse association between PUFA intake and all-cause mortality are consistent with prior evidence [48,50,62]. However, further research is warranted to delve into the effects of PUFA intake on CVD and cancer mortality, given the conflicting results from existing studies [48,50]. The inconsistency in the results may be attributed to the different subtypes of PUFA. A prospective cohort study involving 521,120 participants, which was followed up for 16 years and cited in reference [63], revealed that consuming marine ω-3 polyunsaturated fatty acids in the diet is associated with reduced overall mortality, CVD, and mortality from specific causes. Moreover, studies [64,65] indicate that supplementing the diet with *n*-3 PUFA exhibits notable clinical and statistical significance.

A recent review [66] revealed that the intake of *n*-3 PUFA improves vascular and cardiac hemodynamics, reduces triglycerides, enhances endothelial function, improves autonomic nervous control, alleviates inflammation, decreases thrombosis, and alleviates arrhythmia. This could potentially reduce the risk of death by lowering the risk of inflammation-related diseases, such as CVD and cancer. However, several other studies have examined the association between dietary *n*-6 PUFA and mortality, but the findings are inconsistent, indicating either no association [67,68] or a negative association [28]. Research on the *n*-6/*n*-3 ratio indicates that an increase in the *n*-6/*n*-3 ratio is not associated with an increase in all-cause mortality [28] or a lower all-cause mortality [63]. Moreover, diets high in *n*-6 PUFA may diminish the beneficial impact of *n*-3 PUFA on CHD risk via competitive extension and desaturation pathways [69]. However, due to the diverse potential competition of PUFA for numerous metabolic pathways, it is challenging to predict their impact on CVD risk. Furthermore, an increasing body of research suggests that diets high in *n*-6 PUFA increase the susceptibility of lipids to free radical oxidation and lipid peroxidation, thereby influencing the onset and progression of cancer [70]. More research is needed to determine the differences in dietary PUFA subtypes in order to explain the mechanisms by which different PUFAs affect the occurrence and development of CVD and to provide more accurate references for dietary guidelines.

In relation to MUFA, we discovered that a high intake is inversely associated with both all-cause mortality and specific-cause mortality, but not significantly associated with CVD risk. Unlike our findings, some meta-analyses [48,71] have reported no significant association between mortality from CVD and cancer. Significant heterogeneity exists among studies, attributable in part to inconsistent adjustments for covariates across studies. Another possible reason is that MUFA comes from various food sources, some of which may contain unhealthy nutrients, such as saturated fatty acids or cholesterol, found in meat, dairy products, and partially hydrogenated oils, which may obscure the correlation between total MUFA. Olive oil, nuts, salad dressings, and fried foods primarily contribute to plant-derived MUFA (MUFA-Ps), whereas red and processed meats, dairy products, butter, and poultry are primary sources of animal-derived MUFA (MUFA-As) [72]. The higher the intake of olive oil is, the lower the all-cause and cause-specific mortality is [73,74]. Additionally, higher intakes of nuts are linked to reduced all-cause and specific-cause mortality [27,75,76], whereas processed and red meats are linked to increased mortality [55,56]. Moreover, two prospective cohort studies in the United States have shown that the higher the intake of MUFA-P is, the lower the mortality rate is, while the higher the intake of MUFA-A is, the higher the mortality rate is. The study found that when SFA, refined carbohydrates, or trans fats are substituted with MUFA-P, the mortality rate significantly decreases, while MUFA-A does not [77]. Based on this, we hypothesize that the complex plant-based matrices, including bioactive nutrients and phytochemicals, synergistically stimulate cardiac metabolic pathways [5]. Consequently, this leads to the improved regulation of blood lipids and blood sugar, reduced blood pressure, reduced obesity, and enhanced antioxidant and anti-inflammatory effects, ultimately reducing the risk of CVD, diabetes, and all-cause mortality [78]. However, the majority of studies in the present meta-analysis assessed total MUFA, and the observed variations in results may stem from differences in MUFA sources and subtypes. Further evidence is required to elucidate the differential associations of MUFA-Ps and MUFA-As with variations in CVD incidence and mortality.

### 4.4. Carbohydrate and Health Outcomes

Our research indicates that the association between total carbohydrate intake in the diet and CVD risk differs by gender, but there is no association between dietary carbohydrates and mortality. Furthermore, we observed that as carbohydrate intake increases, both the population as a whole and women experience heightened CVD risk, while no significant association is observed in men. Dose-response analysis further confirms that there is no association between carbohydrate intake and CVD risk.

The long-term health effects of carbohydrate intake in the diet are ambiguous. A meta-analysis of a prospective cohort study involving 15,428 adults revealed a U-shaped association between carbohydrate intake and mortality. Excessive and insufficient carbohydrate intake both increase mortality, with the lowest risk observed at an intake ranging between 50 and 55% [14]. However, another meta-analysis [79] indicated that high-carbohydrate diets are not associated with cardiovascular disease mortality. In 2017, a prospective epidemiological study from 18 countries across five continents [13] found that high carbohydrate intake is associated with increased mortality, but not with the incidence rate of CVD. Inconsistent research results may be attributed to significant variations in the prevalence of various risk factors, dietary components, and carbohydrate quality.

Intake of large amounts of refined carbohydrates, such as white rice, may contribute to various cardiometabolic disorders [80]. Refined carbohydrates can lead to rapid increases in post-meal blood sugar, resulting in a significant release of insulin [81]. This induces the formation of advanced glycation end products, stimulates oxidative stress and inflammatory processes, leads to dyslipidemia, and causes endothelial damage and vascular dysfunction [81,82].

Mounting evidence indicates that high intakes of refined carbohydrates are associated with diabetes [83,84], metabolic syndrome [85], and chronic inflammation [86], all of which are linked to the progression of cardiovascular disease. The observed variations in outcomes between men and women may stem from their distinct metabolic responses to carbohydrates in the diet. The results of two meta-analyses [87,88] indicated that the harmful metabolic effects of a hyperglycemic diet are associated with a higher risk of IHD events in women. Diets high in carbohydrates can reduce the concentration of HDL-C in the bloodstream, thus elevating the risk of coronary heart disease in women, particularly postmenopausal individuals [89]. Concurrently, high-GI diets may elevate TG levels, potentially rendering women more susceptible to CVD compared to men [90].

Given the nuanced relationship between carbohydrate intake and CVD risk, it is crucial to differentiate between simple and complex carbohydrates, as well as the glycemic index of the foods consumed. Therefore, more research is needed to distinguish between simple and complex carbohydrates, as well as the glycemic index of the food consumed, in order to better evaluate the impact of carbohydrates on cardiovascular disease.

### 4.5. Strengths and Weaknesses of the Study

Our research has the following advantages. Firstly, we incorporated a vast number of samples, allowing us to assess the association between the intake of multiple nutrients and the risks of mortality and cardiovascular disease more effectively than any individual study. Secondly, we identified it through a systematic literature search, supplemented by a manual search of published publication reference lists and systematic reviews to ensure no omissions. Thirdly, a dose-response analysis was conducted to evaluate the non-linear associations. Finally, as all included studies were prospective, the impact of recall and selection bias can be ignored. Additionally, we took into account the subtypes of dietary macronutrient intake, encompassing animal and plant-based protein, as well as saturated, monounsaturated, and polyunsaturated fatty acids. In light of this, our study offers a comprehensive understanding of the association between dietary nutrient intake and the risks of mortality and cardiovascular events.

Our meta-analysis also has potential limitations. First, due to the limited research in certain regions, a more detailed sub-ethnic analysis was not feasible, making it impossible to explore the influence of region and culture on diet specifically. This restricts the interpretation of evidence from certain countries or regions. Also, they were included in different cohorts and used different dietary assessment methods, including food frequency questionnaires, dietary recall and recording, as well as different units of constant nutrient intake in different studies. In our meta-analysis, most studies used FFQ (90) or 24 h dietary review (25) to collect dietary indicators, with only seven studies using food records. Although FFQ or 24 h dietary review has become a dietary assessment tool for large-scale nutritional epidemiological studies and has been proven effective in multiple studies, measurement errors in dietary assessment are inevitable. Moreover, it is noteworthy that the included studies in the present analyses used different criteria to define the exposure. These divergent exposure criteria could affect the accuracy of the risk estimates and describe some of the heterogeneity; however, due to the limited number of studies, subgroup analyses could not be confidently conducted to explore the influence of those definitions on our results. Second, most of the studies we included were focused on middle-aged people, and further research is needed on the evidence of the impact of high nutrient intake on the health outcomes of elderly people. Third, we found that most studies only reported total amounts, and few studies distinguished the quality of major nutrients. Further understanding the impact of dietary nutrient quality on diseases will help formulate public health policies and dietary guidelines. Last, although Egger’s test and Begg’s test are the most commonly used methods for testing publication bias, our results also show that no publication bias was found based on either the Begg’s rank correlation test or Egger’s linear regression test in most studies. However, publication bias is not the only source of asymmetry, and there are other sources, including poor method design, insufficient statistical analysis, etc. In addition, this study has a large number of included literature, and some results have heterogeneity. Some of the included studies are small-sample populations, and the selection of more sensitive populations may have an impact on the results of publication bias.

### 4.6. Practical Implications

The intake of macronutrients (protein, fat, and carbohydrates) is undoubtedly important for cardiovascular and overall health, and it is often emphasized in guidelines and clinical practice. However, in the context of an overall healthy eating pattern, balanced nutrient intake is more necessary. This meta-analysis of 124 cohorts suggested that macronutrient takes place as associated with morality and CVD morality, and this effect was different in an analysis structured by the type of macronutrient The impact of balanced nutrient intake on cardiovascular health and mortality is of significant clinical significance, as they may help provide information for developing future dietary strategies to prevent or delay death and provide deeper insights related to cardiovascular health, promoting healthy aging.

However, future research still needs to further elucidate the impact of specific nutritional subtypes on cardiovascular risk factors or disease prevention. Future cohort studies should aim to more accurately and consistently capture the dietary patterns and nutrient intake of different populations. In order to better evaluate the relationship between nutrients and problems and provide accurate references for dietary guidelines, further verification is needed in randomized controlled trials (RCTs). Through the integration and analysis of the results of RCTs, the causal relationship between nutrient intake and cardiovascular outcomes can be established.

## 5. Conclusions

This meta-analysis of 124 cohorts suggests that macronutrient intake is associated with mortality and CVD morbidity, and this effect was different in an analysis stratified by the type of macronutrient. That means balanced nutrient intake is essential in the context of an overall healthy eating pattern. There is a need for further well-designed prospective cohort studies to examine the potential association between macronutrient intake and mortality and CVD morbidity.

## Figures and Tables

**Figure 1 nutrients-16-00152-f001:**
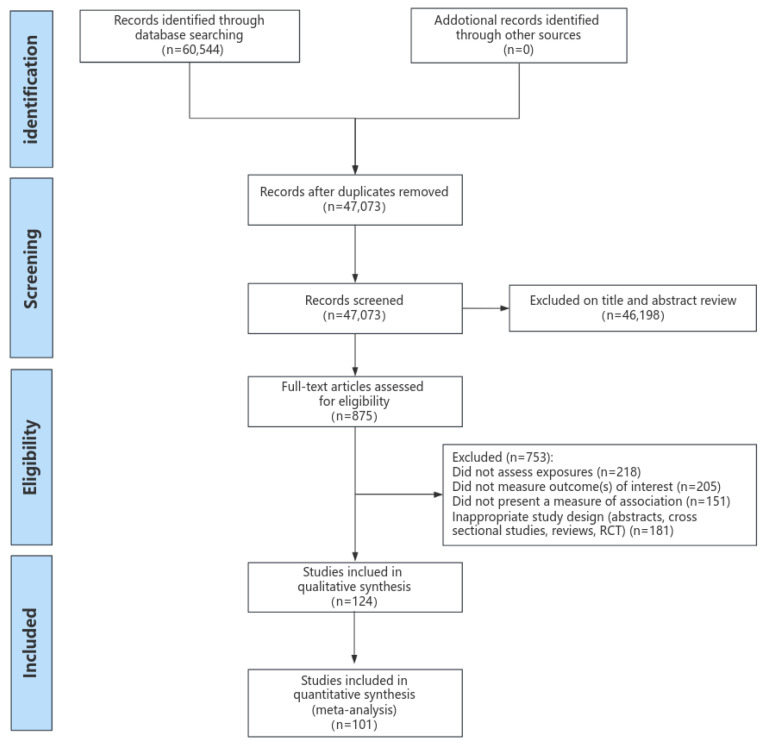
Results of systematic literature search.

**Figure 2 nutrients-16-00152-f002:**
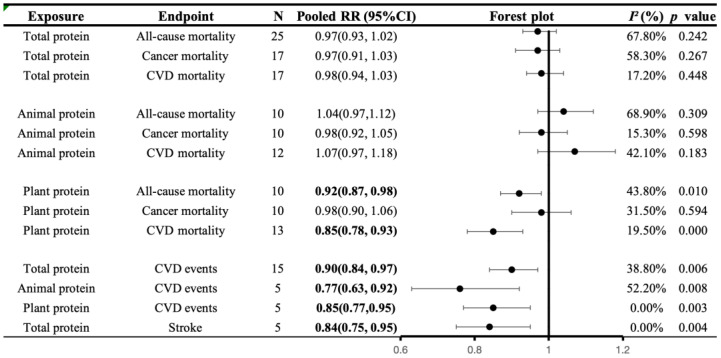
A summary plot for the association of protein intake with CVD morbidity and mortality. Analyses were conducted using generic, inverse variance random-effects models (at least five study comparisons available) or fixed-effects models (fewer than five study comparisons available). Abbreviations: CI, confidence interval; CVD, cardiovascular disease; RR, risk ratio. Statistically significant results will be annotated with bold values.

**Figure 3 nutrients-16-00152-f003:**
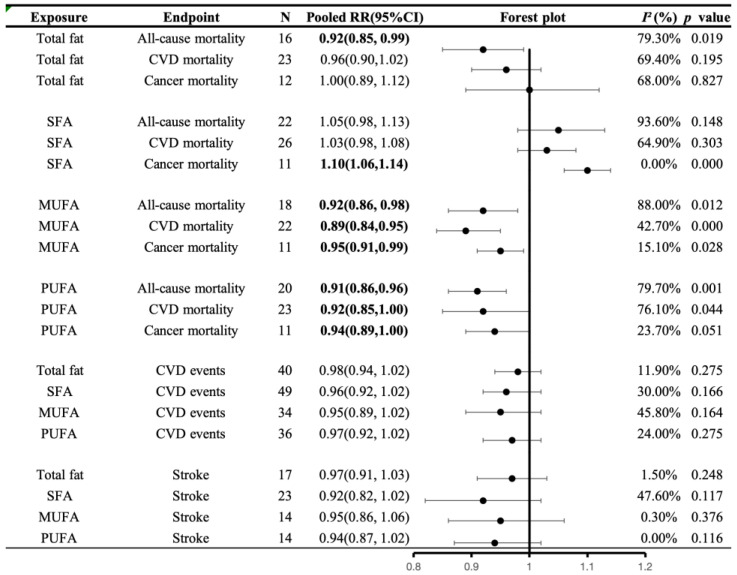
A summary plot for the association of fat intake with CVD morbidity and mortality. Analyses were conducted using generic, inverse variance random-effects models (at least five study comparisons available) or fixed-effects models (fewer than five study comparisons available). Abbreviations: CI, confidence interval; CVD, cardiovascular disease; RR, risk ratio; MUFA, monounsaturated fatty acid; PUFA, polyunsaturated fatty acids; SFA, saturated fatty acid. Statistically significant results will be annotated with bold values.

**Figure 4 nutrients-16-00152-f004:**
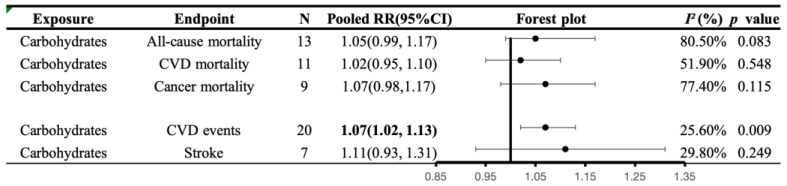
A summary plot for the association of carbohydrates intake with CVD morbidity and mortality. Analyses were conducted using generic, inverse variance random-effects models (at least five study comparisons available) or fixed-effects models (fewer than five study comparisons available). Abbreviations: CI, confidence interval; CVD, cardiovascular disease; RR, risk ratio. Statistically significant results will be annotated with bold values.

**Figure 5 nutrients-16-00152-f005:**
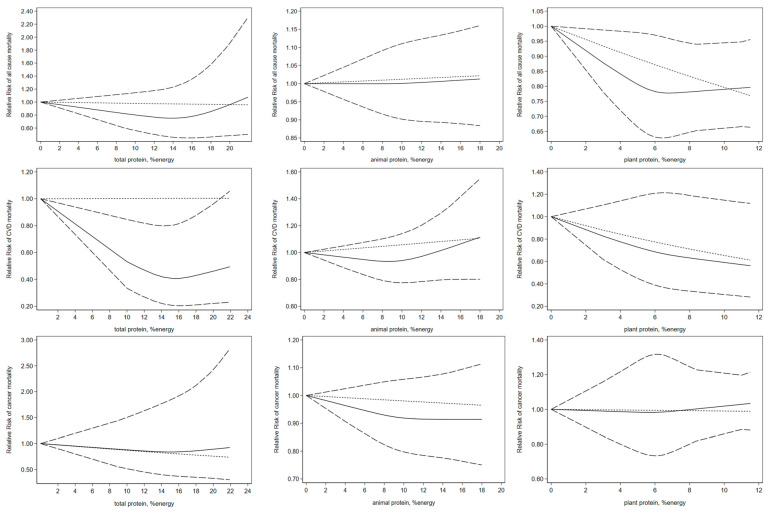
Non-linear dose-response association of intakes of total, animal, plant protein based on percentage of energy with risk of mortality in adults aged 19 or older. Analyses were conducted using random-effects model. Abbreviations: CVD, cardiovascular disease. Solid line represents non-linear dose response and dotted lines represent 95% confidence interval.

**Figure 6 nutrients-16-00152-f006:**
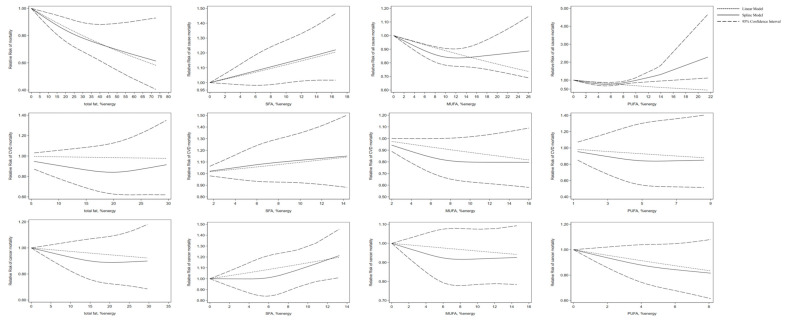
Non-linear dose-response association of intakes of total fat, SFA, MUFA, and PUFA based on percentage of energy with risk of mortality in adults aged 19 or older. Analyses were conducted using a random-effects model. Abbreviations: CVD, cardiovascular disease; MUFA, monounsaturated fatty acid; PUFA, polyunsaturated fatty acids; SFA, saturated fatty acid.

**Figure 7 nutrients-16-00152-f007:**
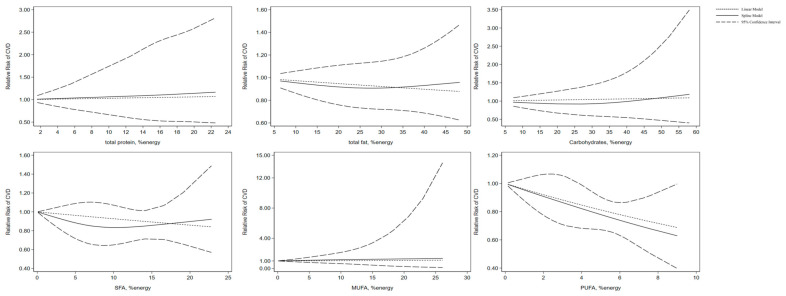
Non-linear dose-response association of intakes of total protein, total fat, carbohydrates, SFA, MUFA, and PUFA based on percentage of energy with risk of CVD in adults aged 19 or older. Analyses were conducted using a random-effects model. Abbreviations: CVD, cardiovascular disease; MUFA, monounsaturated fatty acid; PUFA, polyunsaturated fatty acids; SFA, saturated fatty acid. Solid line represents non-linear dose response and dotted lines represent 95% confidence interval.

**Table 1 nutrients-16-00152-t001:** Summary of the characteristics of the included studies.

Characteristic	All-Cause Mortality and Cause-Specific Mortality			CVD Morbidity		
	Protein	Fats	Carbohydrate	Protein	Fats	Carbohydrate
Number of reports	40	46	19	10	44	9
Study location *	USA (17), Canada (3), Australia (1), Italy (3), Japan (4), Sweden (2), Netherlands (2), UK (2), Bangladesh (1), France (1), Spain (1), Finland (1), China (1)	USA (20), Australia (2), UK (4), Finland (1), Italy (4), Canada (1), Japan (5), Sweden (1), Greece (2), China (3), Korea (2), Bangladesh (1), Norway (1), Spain (1)	USA (4), Sweden (4), Japan (1)	USA (4), Sweden (4), Japan (1)	USA (25), Japan (6), Sweden (5) Finland (2), Iran (2), Greece (2), UK (1), Israel (1), Denmark (1), Finland (1), Norway (1), Spain (1), Holland (1)	USA (6), Sweden (2), Finland (1), China (1), Japan (1)
Sample size (range)	36,035 (278 to 416,104)	39,484 (278 to 521,120)	34,895 (278 to 135,335)	4424 (13 to 135,335)	32,476 (227 to 135,335)	58,191 (3248 to 58,191)
Baseline age (range)	20–74	20–80	20–86 y	40–83 y	34–79 y	38–75 y
Duration (range)	2.2–26 y	4.8–30 y	2.2–26 y	2.5–24.6	4.6–30 y	5.4–30 y
Dietary assessment method	FFQ (28), 24 h recall (10), food record (2)	FFQ (32), 24 h recall (12), food record (2)	FFQ (13), 24 h recall (6)	FFQ (7), 24 h recall (1), record (2)	FFQ (34), 24 h recall (5), record (5)	FFQ (7), 24 h recall (1), record (1)
Outcome assessment method	Confirmed or conducted by investigators (40)	Confirmed or conducted by investigators (46)	Confirmed or conducted by investigators (19)	Conducted by investigators (7), self-reported (3)	Conducted by investigators (25), self-reported (19)	Conducted by investigators (4), self-reported (5)

Data are presented as median (range) or number of cohorts. FFQ: food frequency questionnaire; CVD: Cardiovascular Disease. * Each group includes Dehghan et al., 2017 [13], who included individuals from 18 countries, 3 high-income (Canada, Sweden, and United Arab Emirates), 11 middle-income (Argentina, Brazil, Chile, China, Colombia, Iran, Malaysia, occupied Palestinian territory, Poland, South Africa, and Turkey), and 4 low-income countries (Bangladesh, India, Pakistan, and Zimbabwe).

## Data Availability

All data generated or analyzed during this study are included in this published article (and its Appendix A).

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
