# Peer review of "Dietary Macronutrient Intake and Cardiovascular Disease Risk and Mortality: A Systematic Review and Dose-Response Meta-Analysis of Prospective Cohort Studies"

_nutrients, 2024, doi:10.3390/nu16010152_

Round 1

Reviewer 1 Report

Comments and Suggestions for Authors

Your meta-analysis on macronutrients and cardiovascular health is an impressive contribution to the field. I recommend further detailing on methodological consistency, particularly regarding dietary assessment across studies. The high heterogeneity in the data suggests a need for deeper investigation, possibly through meta-regression techniques. Clarification on the impact of removing influential studies in your sensitivity analysis would be beneficial. Additionally, a more nuanced discussion on the types of proteins, fats, and carbohydrates and their diverse sources could enrich the findings. Given the observational nature of the studies, conclusions should be presented with appropriate caution, emphasizing overall dietary patterns. Lastly, considering the global variation in diets, contextualizing the results for different populations would enhance the manuscript's applicability to global health.

Methodological Rigor

  1. Cohort Selection: The inclusion of over 124 prospective cohort studies is commendable for its robust sample size. However, it is crucial to ensure that the studies are comparable in design, participant characteristics, and outcomes measured.
  2. Nutrient Intake Assessment: Different methods of dietary assessment could introduce significant bias. The use of standardized food frequency questionnaires, 24-hour dietary recalls, or food diaries and the validation of these tools in diverse populations should be discussed.
  3. Adjustments for Confounders: It is vital that the studies included have adjusted for potential confounders such as age, sex, socioeconomic status, other health conditions, and lifestyle factors like smoking and physical activity.

Statistical Analysis

  1. Heterogeneity: High heterogeneity (I²) in some analyses raises questions about the comparability of the studies and the pooled estimates' validity. The sources of heterogeneity should be further explored, possibly through meta-regression.
  2. Sensitivity Analysis: The approach of excluding influential studies is appropriate. However, the impact of these exclusions on the overall results should be discussed more critically, particularly when it leads to significant changes in the risk estimates.
  3. Publication Bias: The use of both Begg and Egger's tests is suitable, but the manuscript should also discuss the limitations of these tests, especially in the context of a large number of studies and the presence of heterogeneity.

Nutrient-Specific Findings

  1. Protein Intake: The manuscript should consider the biological plausibility of the differential effects of animal versus plant proteins on CVD risk and mortality. Furthermore, the impact of protein source and preparation methods should be addressed.
  2. Carbohydrate Intake: Given the nuanced relationship between carbohydrate intake and CVD risk, it is crucial to differentiate between simple and complex carbohydrates, as well as the glycemic index of the foods consumed.
  3. Fat Intake: The manuscript appropriately distinguishes between different types of fats. However, it should delve deeper into the ratio of omega-6 to omega-3 fatty acids and the role of trans fats, which are known to have deleterious cardiovascular effects.

Clinical and Public Health Implications

  1. Dietary Recommendations: The manuscript's conclusions could have significant implications for dietary guidelines. These should be tempered with caution, emphasizing the need for balanced nutrient intake within the context of an overall healthy dietary pattern.
  2. Global Relevance: The findings should be contextualized within global dietary patterns, taking into account regional variations in diet and their implications for CVD risk.

Future Research

  1. Subtype Analysis: More research is needed to assess the impact of specific nutrient subtypes, especially for fats and carbohydrates, on CVD outcomes.
  2. Prospective Cohorts: Future cohort studies should aim to capture dietary patterns and nutrient intake more accurately and consistently across diverse populations.
  3. Randomized Controlled Trials (RCTs): The manuscript would benefit from a discussion on the need for RCTs to establish causal relationships between nutrient intake and cardiovascular outcomes.

Overall Assessment

While the manuscript contributes valuable insights into the associations between macronutrient intake and cardiovascular health outcomes, there is a need for cautious interpretation of the results. The complex nature of diet, lifestyle, and cardiovascular diseases requires a multifaceted approach to public health recommendations, emphasizing whole dietary patterns over individual nutrients.

Author Response

Comments 1: Your meta-analysis on macronutrients and cardiovascular health is an impressive contribution to the field. I recommend further detailing on methodological consistency, particularly regarding dietary assessment across studies. The high heterogeneity in the data suggests a need for deeper investigation, possibly through meta-regression techniques. Clarification on the impact of removing influential studies in your sensitivity analysis would be beneficial. Additionally, a more nuanced discussion on the types of proteins, fats, and carbohydrates and their diverse sources could enrich the findings. Given the observational nature of the studies, conclusions should be presented with appropriate caution, emphasizing overall dietary patterns. Lastly, considering the global variation in diets, contextualizing the results for different populations would enhance the manuscript's applicability to global health.

Response 1: Thank you for your valuable suggestion. We further elaborated on the consistency of methodology in the revised manuscript and added the PECOTS framework for search strategies (Supplementary Table 3), specific NOS scores for included studies (Supplementary Table 8), confounding factor adjustments for each study (Supplementary Table 7), and methods for dietary assessment and outcome indicator measurement for each study (Supplementary Table 6). We discussed and explained the dietary evaluations of different studies in the discussion section. We conducted subgroup analysis to identify potential sources of heterogeneity in some of the results, and performed prior sensitivity analysis by removing each study from the meta-analysis and recalculating the summary association ("leave one method"). We also provided explanations and detailed discussion analysis in the discussion section.

In addition, we have conducted a more detailed discussion on the types and diverse sources of proteins, fats, and carbohydrates, and have responded and explained them one by one in the following comments on the discovery of specific nutrients. At the same time, considering the differences in global diets, the results of different populations are presented in the discussion and background sections to enhance the applicability of the manuscript to global health. Finally, we provide a cautious explanation of the results and emphasize the importance of an overall dietary pattern and balanced nutrition.

Comments 2: Methodological Rigor

1.Cohort Selection: The inclusion of over 124 prospective cohort studies is commendable for its robust sample size. However, it is crucial to ensure that the studies are comparable in design, participant characteristics, and outcomes measured.

2.Nutrient Intake Assessment: Different methods of dietary assessment could introduce significant bias. The use of standardized food frequency questionnaires, 24-hour dietary recalls, or food diaries and the validation of these tools in diverse populations should be discussed.

3.Adjustments for Confounders: It is vital that the studies included have adjusted for potential confounders such as age, sex, socioeconomic status, other health conditions, and lifestyle factors like smoking and physical activity.

Response 2: Thank you for pointing this out. We agree with those comments.

1.     Cohort Selection:

We have conducted a comprehensive and systematic review and evaluation of the studies we have included again to ensure that the studies are comparable in design, participant characteristics, and outcomes measured. All modifications have been highlighted in red

(1) Supplementary Table S3 provides the PECOTS (population, exposure, comparator, outcome, timeline, and setting/study design) . Besides, the included studies are cohort studies with a follow-up time of more than 1 year and a NOS score of 6 or above. We have supplemented the specific NOS scores of each study, as described in Supplementary Table 8 and the Results 3.3. Study Quality Assessment. Supplementary Table S 8 show a summary of the risk of bias (ROS) assessments of the included prospective cohort studies, respectively. Prospective cohort studies as having relatively high quality with NOS 6, based on their risk of bias determinations. Thus, there was no overall concern about the risk of bias in most study comparisons;

All modifications have been marked in red where in the revised manuscript this change can be found – page3, and line 84-93, line 103-106; page 7, and line 170-174.

Thank you again for your careful review.

(2) We have added Table 1 to display the grouping and overall characteristics of the included studies to demonstrate their comparability. The specific basic characteristics of each study are presented in Supplementary Tables 4-5. Considering the impact of regional dietary differences and gender factors on the results, we conducted subgroup analysis to test the robustness of the results. However, due to limited literature and some more specific information not being obtained, subgroup analyses like PUFA subgroup could not be confidently conducted to explore the impact of those definitions on our results.

All modifications have been marked in red where in the revised manuscript this change can be found – page5-6, and line 136-162.

(3) We supplemented the results of the included studies and the methods for measuring dietary indicators and outcomes. Please refer to Supplementary Table 6 for details.The measurement of dietary indicators was mostly done using FFQ or 24-hour dietary review, and the mortality rate was derived from follow-up registration and database links.

All modifications have been marked in red where in the revised manuscript this change can be found – page5, and line 149-151.

2.     Nutrient Intake Assessment:

We have added a discussion section(4.5. Strengths and Weaknesses of the Study) on the use of standardized food frequency questionnaires, 24-hour dietary recall or food diaries, as well as the validation of these tools in different populations. We have also elaborated on this in the limitations section. Besides, we have added measurement methods for dietary indicators included in the study in Table 1 and Supplementary Table 6.

Also, they were included in different cohorts and used different dietary assessment methods, including food frequency questionnaires, dietary recall and recording, as well as different units of constant nutrient intake in different studies. In our meta-analysis, most studies used FFQ (90) or 24-hour dietary review (25) to collect dietary indicators, with only 7 studies using food records. Although FFQ or 24-hour dietary review has become a dietary assessment tool for large-scale nutritional epidemiological studies and has been proven effective in multiple studies, measurement errors in dietary assessment are inevitable. Besieds, it is noteworthy that the included studies in the present analyses used different criteria to define the exposure. These divergent exposure criteria could affect the accuracy of the risk estimates and describe some of the heterogeneity; however, due to the limited number of studies, subgroup analyses could not be confidently conducted to explore the influence of those definitions on our results.

All modifications have been marked in red where in the revised manuscript this change can be found – page 5, and line 149-151, page 21, and line 603-615.

3.     Adjustments for Confounders:

We have added Supplementary Table 7 Confounding variables of included prospective cohort studies. and presented and explained the results(3.3. Study Quality Assessment) accordingly. Most of the studies considered in the analyses controlled for the predetermined primary confounding variable of energy intake. Additionally, 124 studies adjusted for a minimum of four out of six important confounding variables: age, sex, socioeconomic status, other health conditions, and lifestyle factors like smoking and physical activity.

All modifications have been marked in red where in the revised manuscript this change can be found – page 7, and line 164-169.

Comments 3: Statistical Analysis

1.Heterogeneity: High heterogeneity (I²) in some analyses raises questions about the comparability of the studies and the pooled estimates' validity. The sources of heterogeneity should be further explored, possibly through meta-regression.

2.Sensitivity Analysis: The approach of excluding influential studies is appropriate. However, the impact of these exclusions on the overall results should be discussed more critically, particularly when it leads to significant changes in the risk estimates.

3.Publication Bias: The use of both Begg and Egger's tests is suitable, but the manuscript should also discuss the limitations of these tests, especially in the context of a large number of studies and the presence of heterogeneity.

Response 3: Agree. We have, accordingly, modified results and discussion to emphasize this point.

1.     Heterogeneity:

We conducted subgroup analysis and sensitivity analysis to identify potential sources of heterogeneity among some studies, and supplemented the results in the Results and Discussion section of the revised manuscript. For example, in the analysis of animal protein and all-cause mortality, I2 was greater than 50%. When we conducted regional subgroup analysis, we found that heterogeneity in each subgroup was significantly reduced. In addition, the removal of the study by Hern á ndez Alonso et al. and the study by Chen et al. shifted the overall estimate to 1.000 (0.950 to 1.053; I2=35.6% P=0.144) suggested these two studies were influential outliers

All modifications have been marked in red where in the revised manuscript this change can be found – page 16, and line 405-416.

2.     Sensitivity Analysis:

We conducted a prior sensitivity analysis by removing each study from the meta-analysis and recalculating the summary association ("missing one item" method). The pooled RR remains unchanged for most of the results In the revised manuscript, we have added a critical discussion on the impact of these exclusions on the overall results, especially when they lead to significant changes in risk estimates. We have examined these two studies, and the sample size of both studies is relatively large, with the majority being females. In the gender subgroup, the result of a female/male ratio greater than 1 was 0.918 (0.827, 0.986). Gender and sample size may have an impact on the results.

All modifications have been marked in red where in the revised manuscript this change can be found – page 16, and line 405-416.

3.     Publication Bias:

We have added limitations to Begg and Egger's testing in the discussion section of the revised manuscript

Although Egger test and Begg test are the most commonly used methods for testing publication bias. Our results also show that no publication bias was found based on either the Begg rank correlation test or Egger's linear regression test in most studies However, publication bias is not the only source of asymmetry, and there are other sources, including poor method design, insufficient statistical analysis, etc. In addition, this study has a large number of included literature, and some results have heterogeneity. Some of the included studies are small sample populations, and the selection of more sensitive populations may have an impact on the results of publication bias.

All modifications have been marked in red where in the revised manuscript this change can be found – page 21, and line 621-629.

Comments 4: Nutrient-Specific Findings

1.Protein Intake: The manuscript should consider the biological plausibility of the differential effects of animal versus plant proteins on CVD risk and mortality. Furthermore, the impact of protein source and preparation methods should be addressed.

2.Carbohydrate Intake: Given the nuanced relationship between carbohydrate intake and CVD risk, it is crucial to differentiate between simple and complex carbohydrates, as well as the glycemic index of the foods consumed.

3.Fat Intake: The manuscript appropriately distinguishes between different types of fats. However, it should delve deeper into the ratio of omega-6 to omega-3 fatty acids and the role of trans fats, which are known to have deleterious cardiovascular effects.

Response 4: Agree. I/We have, accordingly, done/revised/changed/modified…..to emphasize this point.

1.     Protein Intake:

Regarding the biological validity of the different effects of animal protein and plant protein on cardiovascular disease risk and mortality that you mentioned, we have made modifications and elaborated on the relationship between some proteins and health outcomes in the discussion section.

All modifications have been marked in red where in the revised manuscript this change can be found – page 17, and line 447-451, line 458-465.

2.     Carbohydrate Intake:

Given the subtle relationship between carbohydrate intake and cardiovascular disease risk, distinguishing between simple carbohydrates and complex carbohydrates, as well as the glycemic index of the food consumed, is crucial. However, due to the limited quantity of carbohydrates included in the study and the lack of differentiation between simple carbohydrates and complex compounds in relevant studies, we were unable to conduct such assessments, But we discussed refined carbohydrates and glycemic index in the discussion section, and looked forward to future research. Therefore, more research is needed to distinguish between simple and complex carbohydrates, as well as the glycemic index of the food consumed, in order to better evaluate the impact of carbohydrates on cardiovascular disease.

All modifications have been marked in red where in the revised manuscript this change can be found – page 20, and line 563-584.

3.     Fat Intake:

In this study, we evaluated the association between SF, MUFA, and PUFA with cardiovascular disease and mortality. You mentioned that the role of omega-6 and omega-3 fatty acid ratios should be further studied. Therefore, we conducted a detailed screening of the included studies. However, due to the limited proportion of omega-6 and omega-3 fatty acids involved in the included studies, we were unable to conduct a more detailed subgroup analysis, However, we will conduct relevant discussions and explanations in the discussion section. In addition, considering that fats have a clear cardiovascular risk and that most studies have not included trans fats in the assessment of fat intake, we excluded studies containing trans fats during the screening stage and did not include them in the meta-analysis results.

All modifications have been marked in red where in the revised manuscript this change can be found – page18- 19, and line 484-487, line 512-516, line 541-544.

Comments 5: Clinical and Public Health Implications

1.Dietary Recommendations: The manuscript's conclusions could have significant implications for dietary guidelines. These should be tempered with caution, emphasizing the need for balanced nutrient intake within the context of an overall healthy dietary pattern.

2.Global Relevance: The findings should be contextualized within global dietary patterns, taking into account regional variations in diet and their implications for CVD risk.

Response 5: Agree. We have, accordingly, modified Discussion to emphasize this point.

1.Dietary Recommendations:

We emphasized the need for balanced nutrition within the context of an overall health dietary pattern in the discussion section, 4.6. Practical Implications.

All modifications have been marked in red where in the revised manuscript this change can be found – page 22, and line 630-649.

2.Global Relevance:

We have added a discussion section that combines global dietary patterns and considers regional differences in diet and its impact on cardiovascular disease risk.

All modifications have been marked in red where in the revised manuscript this change can be found – page 21, and line 599-603.

Comments 6:Future Research

1.Subtype Analysis: More research is needed to assess the impact of specific nutrient subtypes, especially for fats and carbohydrates, on CVD outcomes.

2.Prospective Cohorts: Future cohort studies should aim to capture dietary patterns and nutrient intake more accurately and consistently across diverse populations.

3.Randomized Controlled Trials (RCTs): The manuscript would benefit from a discussion on the need for RCTs to establish causal relationships between nutrient intake and cardiovascular

Response 6: Agree. We greatly appreciate the comments and suggested revisions to our paper. We emphasized this section in the discussion section

1.     Subtype Analysis:

We have made corresponding modifications to this suggestion in the final sections of protein and health outcomes, fat and health outcomes, and carbohydrates and health outcomes, emphasizing the need for more detailed research to evaluate the impact of specific nutrient subtypes on cardiovascular disease and mortality.

All modifications have been marked in red where in the revised manuscript this change can be found – page 17, and line 447-451, line 458-465, page18- 19, and line 484-487, line 512-516, line 541-544, page 20, and line 563-584.

2.Prospective Cohorts:

In addition to emphasizing the discussion of health outcomes mentioned above, we once again look forward to future cohort studies that need to capture the dietary patterns and nutrient intake of different populations more accurately and consistently, in order to better evaluate the association between nutrients and diseases and provide correct references for dietary guidelines.

All modifications have been marked in red where in the revised manuscript this change can be found – page 21, and line 641-649.

3.Randomized Controlled Trials (RCTs):

Randomized controlled trials, as a high level of evidence, play an important role in exploring the causal relationship between dietary nutrients and cardiovascular disease. Therefore, further validation is needed in randomized controlled trials in the future, and the results of RCTs need to be integrated and analyzed. You can find this modification in the discussion section of the revised manuscript

All modifications have been marked in red where in the revised manuscript this change can be found – page 21, and line 641-649.

Comments 7: Overall Assessment

 While the manuscript contributes valuable insights into the associations between macronutrient intake and cardiovascular health outcomes, there is a need for cautious interpretation of the results. The complex nature of diet, lifestyle, and cardiovascular diseases requires a multifaceted approach to public health recommendations, emphasizing whole dietary patterns over individual nutrients.

Response 7: Thank you for your valuable suggestions and comments. We have made corresponding modifications in response to your mentioned modification suggestions, supplementing and modifying the rigor of the method, statistical analysis, specific nutrient discovery, clinical and public health impact, and future research. At the same time, we have interpreted the results cautiously, emphasizing the overall dietary pattern rather than individual nutrients.

Reviewer 2 Report

Comments and Suggestions for Authors

The authors performed a systematic review and dose-response meta-analysis of prospectove studies on the association of macronuntrients (lipid, carbohydrates, protein) and total, cardiovascular and cancer mortality as well as incidence of cardiovascular events and stroke.

Major problem is data presentation and interpretation. There are numerous discrepancies between the text in the results section and the data in the presented figures.

Several associations with p less than 0.05 are presented as non-significant and vice-versa.

Figures are of poor quality, font too small and the title and legend to the figures is missing.

The Results section of the manuscript should be completely re-written.

Comments on the Quality of English Language

Minor editing.

Author Response

Comments 1: The authors performed a systematic review and dose-response meta-analysis of prospectove studies on the association of macronuntrients (lipid, carbohydrates, protein) and total, cardiovascular and cancer mortality as well as incidence of cardiovascular events and stroke.

Major problem is data presentation and interpretation. There are numerous discrepancies between the text in the results section and the data in the presented figures.

Several associations with p less than 0.05 are presented as non-significant and vice-versa.

Figures are of poor quality, font too small and the title and legend to the figures is missing.

The Results section of the manuscript should be completely re-written.

Response 1: Thank you for pointing this out. In response to the inumerous discrepancies between the text in the results section and the data in the presented figures, we conducted a thorough investigation of all the data and reanalyzed the relevant indicators, with the results section rewritten. In addition, the legend of the chart has been supplemented and improved as required. All modifications have been marked in red where in the revised manuscript this change can be found – page 20-28, and line 210-423.

Thank you again for your careful review.

4. Response to Comments on the Quality of English Language

Point 1: Minor editing.

Response 1:  Thank you for your consideration of our manuscript. We have improved the grammar and edited the manuscript carefully

Round 2

Reviewer 2 Report

Comments and Suggestions for Authors

The manuscript is substantially improved.

Comments on the Quality of English Language

Moderate editing required.

Author Response

We greatly appreciate comments and suggestions for modifications to our document. In this revision, we have improved the grammar and carefully edited the manuscript. We hope that these revisions will lead to the final acceptance of the manuscript for publication. Thank you for your consideration of our manuscript.